# Resistome, Mobilome and Virulome Analysis of *Shewanella*
*algae* and *Vibrio* spp. Strains Isolated in Italian Aquaculture Centers

**DOI:** 10.3390/microorganisms8040572

**Published:** 2020-04-15

**Authors:** Vanessa Zago, Laura Veschetti, Cristina Patuzzo, Giovanni Malerba, Maria M. Lleo

**Affiliations:** 1Department of Diagnostics and Public Health, University of Verona, Strada Le Grazie 8, 37134 Verona, Italy; vanessa.zago@hotmail.com; 2Department of Neurosciences, Biomedicine and Movement Sciences, University of Verona, Strada Le Grazie 8, 37134 Verona, Italy; laura.veschetti@univr.it (L.V.); cristina.patuzzo@univr.it (C.P.); giovanni.malerba@univr.it (G.M.)

**Keywords:** whole genome sequencing, antimicrobial resistance, shewanellae, vibrios, aquatic environment, public health

## Abstract

Antimicrobial resistance is a major public health concern restricted not only to healthcare settings but also to veterinary and environmental ones. In this study, we analyzed, by whole genome sequencing (WGS) the resistome, mobilome and virulome of 12 multidrug-resistant (MDR) marine strains belonging to *Shewanellaceae* and *Vibrionaceae* families collected at aquaculture centers in Italy. The results evidenced the presence of several resistance mechanisms including enzyme and efflux pump systems conferring resistance to beta-lactams, quinolones, tetracyclines, macrolides, polymyxins, chloramphenicol, fosfomycin, erythromycin, detergents and heavy metals. Mobilome analysis did not find circular elements but class I integrons, integrative and conjugative element (ICE) associated modules, prophages and different insertion sequence (IS) family transposases. These mobile genetic elements (MGEs) are usually present in other aquatic bacteria but also in *Enterobacteriaceae* suggesting their transferability among autochthonous and allochthonous bacteria of the resilient microbiota. Regarding the presence of virulence factors, hemolytic activity was detected both in the *Shewanella algae* and in *Vibrio* spp. strains. To conclude, these data indicate the role as a reservoir of resistance and virulence genes in the environment of the aquatic microbiota present in the examined Italian fish farms that potentially might be transferred to bacteria of medical interest.

## 1. Introduction

Shewanellae are non-fermentative Gram-negative, motile rods that inhabit aquatic and sedimentary environments. Cases of human infections involving *Shewanella* are rare but may include skin and soft tissue infections, septicemia, hepatobiliary disease, otitis media and pneumonia. These infections are described in immunocompromised patients with renal failure, neutropenia, hepatobiliary disease, diabetes or those involved in trauma accidents [1,2,3,4,5]. Usually, the antibiotic therapy adopted includes beta-lactams, aminoglycosides and quinolones. These bacteria are generally susceptible to third and fourth generation cephalosporins, carbapenems, beta-lactamase inhibitor combinations, aminoglycosides, chloramphenicol, erythromycin, aztreonam and quinolones [2,5,6]. However, resistance to these drugs is increasing due to the presence in their chromosome of class D beta-lactamase encoding genes (*bla_OXA_*) conferring resistance to carbapenems, class C beta-lactamases (*bla_Amp_*_C_) which decrease the susceptibility to cephalosporins and *qnr* genes responsible for resistance to quinolones [4,7,8,9,10,11,12,13,14]. Furthermore, resistance to colistin, currently a last-resort antibiotic in human medicine, has been reported as well, due to the presence of the chromosomal *eptA* gene encoding for phosphoethanolamine transferase [12,15,16]. Regarding virulence-associated factors, hemolytic activity has been reported [12]. Also vibrios are included in the marine microbial community and they are able to cause severe or mild gastroenteritis in humans [17,18]. Most of the species belonging to the genus *Vibrio* are harmless to humans except for the “big four” representing by *V. cholerae*, *V. parahaemolyticus*, *V. vulnificus* and *V. alginolyticus* [19]. All of them are waterborne and foodborne pathogens. *V. cholerae*, *V. parahaemolyticus* but also other environmental vibrios may contribute to the spread of resistance through mobile genetic elements (MGEs) [20]. *V. anguillarum* represents the causative agent of a deadly haemorrhagic septicaemic disease affecting various marine and fresh or brackish water fish, bivalves and crustaceans [21]. It harbors a genome containing several MGEs such as plasmids, superintegrons and insertion sequences (ISs) carrying cargo genes, particularly resistance and virulence genes [21]. Tetracyclines, aminoglycosides, third generation cephalosporins and fluoroquinolones are generally used as antibiotic therapy in human vibriosis outbreaks [13,22] but, several cases of multidrug-resistant (MDR) vibrios are reported in literature even to the last-resort antibiotics used in human medicine such as carbapenems [22,23,24,25,26,27].

Another mechanism of resistance that can be detected, using whole genome sequencing (WGS), is the presence of multidrug efflux pump systems. They represent the first line of defense against antibiotics as these pumps decrease the intracellular level of drugs while the bacterial cell activates other levels of protection such as the production of enzymes [28]. Bacterial MDR efflux pumps are classified into five different structural families: the adenosine triphosphate (ATP)-binding cassette (ABC) superfamily [29], the multidrug and toxic compound extrusion (MATE) family [30], the major facilitator superfamily (MFS) [31], the small multidrug resistance (SMR) family [32], and the resistance/nodulation/division (RND) superfamily [33]. Their mechanism of action relies on exporting the drug out of the cell and are very common in Gram negative bacteria included *Shewanellaceae* and *Vibrionaceae* strains. As an example, the detection of *mexF* gene in *Shewanella* is required for resistance to antibiotics such as chloramphenicol and tetracycline [34]. Integrative and conjugative elements (ICEs), belonging in particular to SXT/R391 family and carrying genes involved in antimicrobial resistance (AMR), virulence and heavy metal resistance, are the main mobile elements found in these aquatic bacteria which are able to contribute to the spread of these genes in the environment [35,36,37]. Moreover, also class I integron and other transposable elements (insertion sequences) have been detected in these two genera [21,38,39,40].

Next-generation sequencing (NGS) is becoming a powerful tool in microbiology facilitating the detection of AMR and virulence factors encoding genes as well as those associated to their genetic mechanisms [41]. In particular, WGS has already numerous applications in AMR ranging from the development of novel antibiotics, surveillance systems both in human and veterinary medicine, the study of the evolution of resistance in real-time under a variety of conditions to the development of diagnostic tests [42]. In this work, we used this technology to investigate the resistome, mobilome and virulome of 12 MDR *S. algae* and *Vibrio* spp. strains isolated in several Italian aquaculture centers in order to study the resistance mechanisms involved, their contribution as antimicrobial resistance gene (ARG) reservoir and the possible outcomes in public health.

## 2. Materials and Methods

### 2.1. Strain Identification, Phenotypical and Biochemical Analysis

Twelve MDR marine bacteria belonging to a collection of *Shewanellaceae* and *Vibrionaceae* strains isolated in fish farms located along the coast of the Adriatic Sea, in Italy [43], were selected for the WGS analysis. The selection was based on their resistance (antimicrobial susceptibility testing) and biochemical profiles (positiveness to Blue Carba Test). Growth on ChromID^®^ extended spectrum beta-lactamase (ESBL) medium (Biomérieux, Marcy l’Étoile, France) was tested in order to detect ESBL-producing strains. The Blue Carba Test (BCT) [44] was used to detect the presence of carbapenemases through imipenem hydrolysis. Carbapenemase activity was revealed when the test and the negative-control solutions were green versus blue, respectively. Non-carbapenemase producers remained blue or green on both solutions. Bacterial growth was carried out overnight on trypticase soy agar (TSA; Oxoid Ltd., Basingstoke, Hampshire, England) supplemented with NaCl 1% at 37 °C for shewanellae whereas the growth was sustained on TCBS (Oxoid Ltd., Basingstoke, Hampshire, England) at 37 °C for *V. parahaemolyticus* VPE116 and at room temperature for *V. anguillarum* 28AD. Species identification was confirmed using BLAST analysis of the predicted 16S rRNA gene sequence.

### 2.2. Antimicrobial Susceptibility Testing

Antimicrobial susceptibility testing was performed using disk diffusion method in Mueller Hinton (MH) agar plates (Bio-Rad, Cressier Switzerland) according to EUCAST guidelines. The antibiotics used in this study were at following concentrations: amoxicillin (25 µg/mL), amoxicillin+clavulanic acid (30 µg/mL), ticarcillin (75 µg/mL), ticarcillin+clavulanic acid (85 µg/mL), piperacillin (75 µg/mL), piperacillin+tazobactam (85 µg/mL), ceftazidime (10 µg/mL), cefepime (30 µg/mL), temocillin (30 µg/mL), cefoxitin (30 µg/mL), cefotaxime (30 µg/mL), cefepime (30 µg/mL), cephalothin (30 µg/mL), aztreonam (30 µg/mL), imipenem (10 µg/mL), ertapenem (10 µg/mL), meropenem (10 µg/mL), trimethoprim+sulfamethoxazole (25 µg/mL), colistin (50 µg/mL), chloramphenicol (30 µg/mL), sulfonamides (200 µg/mL), fosfomycin (200 µg/mL), tobramycin (10 µg/mL), kanamycin (30 µg/mL), gentamicin (15 µg/mL), amikacin (30 µg/mL), nalidixic acid (30 µg/mL), ciprofloxacin (5 µg/mL), tetracycline (30 µg/mL), and tigecycline (15 µg/mL).

### 2.3. Genomic DNA Extraction and Genome Sequencing

Genomic DNA (gDNA) was extracted using the CTAB method and 1 µg of gDNA was used by Illumina TruSeq DNA PCR-free kit (Illumina, Milan, Italy) for library preparation. WGS was performed by Illumina NextSeq^®^ 500 platform (Illumina, San Diego, CA, USA). It enabled the sequencing of 150 bp paired-end DNA reads using an average sequencing coverage of 400X. Reads underwent a quality control that included an inspection of the overall sequencing quality (FastQC software; https://www.bioinformatics.babraham.ac.uk/projects/fastqc/), trimming of the low-quality sequences and removal of the adapter sequences using Scythe v0.991 (https://github.com/ucdavis-bioinformatics/scythe) and Sickle v1.33 softwares (https://codeload.github.com/najoshi/sickle/tar.gz/v1.33). Then reads were assembled into several genome contigs by SPAdes v3.10.1 assembler [45]. FASTQ reads from all sequences and strains were submitted to the European Nucleotide Archive (https://www.ebi.ac.uk/ena) under BioProject number PRJEB36298.

### 2.4. Genome Annotation

Genome annotation was carried out using Prodigal v2.6, a tool provided by PROKKA v1.12 software [46]. Antimicrobial resistance genes were detected using several databases such as ResFinder, the Comprehensive Antibiotic Resistance Database (CARD), ABRicate and Antibiotic Resistance Gene ANNOTation (ARG-ANNOT) [47,48,49]. Plasmids and other mobile genetic elements such as integrons and transposons were investigated using INTEGRALL [50] and ISFinder databases [51], whereas VR profile v2.0 and ICEBerg v2.0 were used for ICE detection [52]. Finally the PHASTER web server was used for the detection of phages [53] whereas Recycler tool [54] for plasmids. Virulence factors were detected using gene annotation and prediction performed by Prodigal.

## 3. Results

### 3.1. Bacterial Strain Characterization, Phenotypical and Biochemical Results

All the marine strains were identified at species level by WGS analysis blasting the predicted 16S rRNA gene. The 12 strains, under study, were identified as *S. algae* (*n* = 10), *V. anguillarum* (*n* = 1) and *V. parahaemolyticus* (*n* = 1) as reported in Table 1.

The BCT resulted positive for every *S. algae* isolate. Carbapenemase activity was revealed for the 10 *S. algae* strains whereas a doubtful result was obtained for *V. anguillarum* 28AD. The strain VPE116 resulted negative (blue versus blue) and it was used as a negative control (non-carbapenemase producing strain). Moreover, strain growth was inhibited on ChromID^®^ ESBL medium indicating the susceptibility to third generation cephalosporins and the absence of ESBLs.

### 3.2. Antimicrobial Susceptibility Testing

Antimicrobial resistance profiles were obtained by standard antimicrobial susceptibility testing performed according to EUCAST guidelines. As regards the *S. algae* isolates, resistance towards β-lactams (including imipenem), colistin, sulfonamides and fosfomycin was reported (Table 1).

Only the strains 57CP and 38LV were resistant also to tetracycline and tigecycline. *V. anguillarum* 28AD was resistant to the same antibiotics detected in all the *S. algae* strains including temocillin, amikacin, ticarcillin alone and combined with clavulanic acid as well. *V. parahaemolyticus* VPE116 showed resistance to aminoglycosides (tobramycin, kanamycin, amikacin and gentamicin), colistin and sulfonamides but susceptible to imipenem, cephalothin and cefotaxime.

### 3.3. Genome Features

Genome sizes, total number of the generated reads, contigs, predicted genes, proteins, hypothetical proteins, RNA encoding genes and %GC contents are reported in Appendix A (Appendix A) for each isolates. A k-mer value of 97 was selected to obtain the best genome assembly for all the strains.

### 3.4. Resistome Analysis

The resistome analysis evidenced resistance to several antibiotic classes including beta-lactams, quinolones, tetracyclines, macrolides, polymyxins, chloramphenicol and heavy metals.

Resistance to beta-lactams was mainly due to the presence in *S. algae* strains of *bla*_OXA-55-like_, *bla*_AmpC_ and *mexB-OprM* genes which conferred resistance to carbapenems, cephalosporins and penicillins. A *bla*_AmpC_ encoding gene was detected in *V. anguillarum* 28AD whereas *mexA-OprM* and *bla*_CARB-19_ were found in *V. parahaemolyticus* VPE116 (Table 2).

These chromosome-encoded beta-lactamases are regulated by different transcriptional regulators. In particular, in all *S. algae* strains the *bla*_OXA-55-like_ expression is regulated by a member of LysR family transcriptional regulator as already reported in literature [55]. Concerning *bla*_AmpC_, it is located near *cat* (chloramphenicol acetyltransferase), HTH-type transcriptional regulator DmlR, *czcA* (cobalt-zinc-cadmium resistance protein), *OprM* (Outer membrane protein OprM) and *pqiAB* (paraquat-inducible protein AB) genes (Figure 1).

AmpC induction is mediated by beta-lactams and regulated by AmpD, AmpE, AmpR, a LysR-type regulator and AmpG, a series of genes not always found in our sequences. This is probably due to the difficulties encountered in short read sequencing. As regards the *bla*_AmpC_ detected in *V. anguillarum* 28AD isolate, it is regulated by the same system described above but the *cat* gene, although not involved in the beta-lactamase regulation, is not present near the *bla*_AmpC_ gene. Only its activator AmpR has been detected. By contrast, strain VPE116 harbored the *bla*_CARB-19_ gene which was located near the *cusAB* genes involved in silver, copper and fosfomycin resistance.

Concerning the resistance to quinolones, QnrVC6, an integron-mediated quinolone resistance protein, was found only in *V. anguillarum* 28AD isolate, while QnrA7, a plasmid-mediated quinolone resistance (PMQR) protein found in *S. algae*, was present only in some of the shewanellae strains here studied (Table 2). The isolates 353M, 219VB, 38LV, 57CP and 60CP, instead, resulted positive for the presence of the *qnrA3* gene encoding a quinolone-resistant protein reported also in *Escherichia coli* and *Salmonella enteritidis* [56,57]. Mutations on gyrase A and ParC (topoisomerase IV) genes were found in quinolone resistance determining regions (QRDRs). However, the most important mutations, Ser83 and Asp87, were conserved in all the strains in gyrA but not in parC protein sequences where a Ser83Pro was detected. Conversely, the gyrB and parE protein sequences did not show mutations of particular interest in QRDRs.

Several efflux pump families were detected in our *S. algae* and *Vibrio* spp. strains (Table 2).

Quinolone resistance can be mediated by efflux mechanisms using efflux pumps belonging to the MATE and MFS families. In particular, MdtK, EmrAB-TolC and MepAB were present in all the analyzed strains, whereas NorM, Bmr3, MdtH and MfpA were found only in the vibrio isolates (Table 2). Moreover, the regulation of the *EmrAB-TolC* system was described by the presence of an HTH-type transcriptional regulator DmlR (belonging to LysR family transcriptional regulator) which could regulate the EmrAB expression. The DmlR regulator is followed by an acetyltransferase belonging to the GNAT family N-acetyltransferase, which could contribute to quinolone resistance in the *S. algae* isolates.

A list of an important multidrug resistance efflux pump system belonging to RND family conferring resistance to a variety of molecules was identified for each strain analyzed and reported in Table 2. Among them, AcrAB-TolC, AcrE-TolC, MdtAC-TolC, EmrD (MFS family) and MdtN were found in all the *S. algae* strains. EmrYK-TolC, a member of the MFS family, was found only in some *S. algae* (144bCP, 178CP, 146bCP, 83CP, 57CP, 60CP) and in the two vibrios. In addition, *V. parahaemolyticus* VPE116 harbored the *mdtG* (MFS family) and *mdlB* (ABC family) genes contributing to the generation of a resistant phenotype against fluoroquinolones and fosfomycin, respectively.

As concerns pumps involved in tetracycline resistance, *tet34*, *tet35* as well as *tetR*, their regulator, were detected only in the two vibrios. Finally, an ABC-type tripartite multidrug efflux pump, MacAB-TolC, was responsible for macrolide antibiotic resistance. Also MdtE-TolC, a RND-type efflux pump, contributed to macrolide and beta-lactam resistance. The first was detected in all the studied strains while the second one was found in the vibrio genomes but not in the *S. algae* ones (Table 2).

Phosphoethanolamine transferase (*eptA*), a chromosomal encoding gene, was detected in both the two genera causing a resistant phenotype towards polymyxins. Moreover, *cat* and *mdtL* genes, involved in the chloramphenicol resistance, were found.

Resistance to heavy metal was also observed (Appendix A). Some of the genes were found in all the 12 analyzed strains, namely all the genes involved in resistance to arsenic, copper, molybdenum and the *czcD* (cobalt-zinc-cadmium resistance protein), *corC* and *mgtE* genes involved in magnesium resistance. Others were typical of vibrios such as *czcR*, *zur*, *znuA*, *znuB*, *znuC* (cobalt, zinc and cadmium resistance) or of *S. algae*, *chrA* for chromium, *corA* for magnesium and *nikR* for nickel resistance.

### 3.5. Mobilome Analysis

The MGE analysis has revealed the absence of circular elements such as plasmids. Anyway, ICEs, class I integrons, ISs and bacteriophages were detected. Of interest was the presence in 353M, a strain isolated in the open sea, of several genetic elements associated to *int-xis* (integration and excision module), *mob* (DNA mobilization and processing module) and *mpf* (mating-pair formation module) machineries that are typical of ICEs. The isolate 353M carried the following genes: *traID*, *traC*, *traN*, *traG*, a tyrosine-type XerD recombinase (integrase), a site-specific integrase and several other conjugal transfer proteins predicted as hypothetical proteins (Figure 2) but containing protein domains involved in the conjugative process as confirmed by BLAST analysis.

In particular, the Tra amino acid sequences showed high similiarity (99–100%) to *Salmonella enterica* ICE transfer apparatus (TraC, TraG, TraI, TraN) whereas TraD revealed 61.71% of similarity to the same protein detected in *Pseudoalteromonas* sp. GutCa3 (WP_101217907). Concerning the integration module, the site-specific integrase sequence protein was 89.09% similar to that found in *S. putrefaciens* (WP_086903394.1) whereas the XerD recombinase showed low similarity (41.03%) to the tyrosine-type recombinase/integrase found in the marine bacterium *Halofilum ochraceum* (WP_070988554.1). Furthermore, several genes involved in the partitioning such as ParAB, replication (RepA, DNA polymerase III, DNA topoisomerase III, DNA binding proteins, IncW-like replication protein), recombination (RecF, SbcC) and several other genetic elements associated to type II, III and IV secretion systems were revealed. In addition, in 8 out of 10 *S. algae* strains, we found a 200 kbp genetic element containing metabolism, virulence factor, antibiotic and heavy metal resistance encoding genes. Most of these resistance genes were efflux pumps. They were also flanked by integration and recombination genes such as integrases, endonucleases, transposases and phage-associated elements. They seem to be associated to ICE elements considering their large size and the presence of the genes mentioned above. To confirm this, we have also found in the surrounding of these elements a tRNA^Phe^ encoding gene as a possible insertion site in the host chromosome. Moreover, an integration host factor (*ihf*) gene was found to flank this region with the alfa subunit near to the tRNA^Phe^ gene and the beta subunit at the opposite end of this genetic element. Conversely, these elements were absent in the vibrios under study.

Class I integron elements were detected in all the *S. algae* and in *V. anguillarum* 28AD but not in *V. parahaemolyticus* VPE116 in which only an *IntI* was found. *IntI* was also detected in 5 out of 10 *S. algae* strains. Regarding ISs, several families already described in *Enterobacteriaceae* and in other aquatic bacteria such as *Aeromonas salmonicida*, *V. vulnificus*, *S. loihica*, *S. oneidensis*, *S. baltica*, *V. anguillarum*, *V. parahaemolyticus*, *V. splendidus* and *S. putrefaciens* were present in the *S. algae* or *Vibrio* spp. isolates (Table 3).

The presence of bacteriophages and their related elements was investigated in this study as well. As reported in Table 3, several prophages were detected also if incomplete. Interestingly, the presence of phages associated to other aquatic bacteria such as *Aeromonas* phage phiO18P (NC_009542) and *Shewanella* sp. phage 1/44 (NC_025463) was shown in the *S. algae* strains. Moreover, phages derived from *Enterobacteriaceae* family members such as *Escherichia* phage D108 (NC_013594), *Enterobacteria* phage phi92 (NC_023693) and *Enterobacterial* phage mEp213 (NC_019720) were detected in the *S. algae* and in *V. parahaemolyticus* VPE116 isolates. *Vibrio* phage VP882 (NC_009016) and bacteriophage phi 1.45 were found only in *V. parahaemolyticus* VPE116 whereas the strain 38LV did not harbor any phages. In addition to these elements, the inovirus Gp2 encoding gene was found in 6 out of 12 strains (144bCP, 146bCP, 178CP, 38LV, 83CP, 28AD). This protein is associated to a genus of viruses able to infect Gram-negative and Gram-positive bacteria. In addition, a defense system from foreign DNA, the CRISPR-Cas system, was detected in 3 out of 10 *S. algae* strains (219VB, 38LV, 82CP) and in *V. parahaemolyticus* VPE116 isolate.

### 3.6. Virulome Analysis

Virulence factor encoding genes were investigated both in the *S. algae* and *Vibrio* spp. isolates (Appendix A). Hemolytic activity is predominant in these strains. In particular, hlyD, hlyD family secretion protein and hemolysin III were all found in the *S. algae* strains as well as a complex machinery for the secretion of these factors such as type I, II, III, IV and VI secretion systems. Regarding *V. anguillarum* 28AD, a thermolabile hemolysin, hemolysin III, hemagglutinin, the repeats-in-toxin subunit A (rtxA) and type I, II, III, IV and VI secretion systems were found. In addition to these factors, *V. parahaemolyticus* VPE116 harbored hemolysin D and the thermostable hemolysin delta-VPH as well.

### 3.7. Bacteriocins

Furthermore, some bacteriocin encoding genes such as *lod*A, L-lysine 6-oxidase, *lod*B, a putative FAD-dependent oxidoreductase and *cvp*A, a colicin V production protein were detected in all the *S. algae* strains. In *V. anguillarum* 28AD, the expressing hydrolase activity gene *vabF* was detected.

## 4. Discussion

Antibiotics are widely used in human medicine, food producing animals and agricultural activities to prevent and treat infections. In the aquaculture sector, a lack of legislation has led to an indiscriminate use of antibiotics which may spread in the aquatic environment and sediment promoting antimicrobial resistance in the aquatic microbiota [43,59]. Therefore, these bacteria represent a potential reservoir of ARGs [60] and a risk to public health [61]. The classes of antibiotics used both for therapeutic and prophylaxis, on the basis of legal permissions in Italian aquaculture, include beta-lactams (amoxicillin, ampicillin), tetracyclines (clortetracycline, tetracycline, oxytetracycline), amphenicols (florfenicol, thiamphenicol), macrolides (erythromycin), sulfonamides (all, included trimethoprim+sulfonamide) and quinolones (oxolinic acid, flumequin, enrofloxacin) [62,63]. A review of the scientific literature on AMR in Italian aquaculture has evidenced a common resistance to beta-lactams, tetracycline, erythromycin and trimethoprim, and a general susceptibility to chloramphenicol, tobramycin and flumequin [64]. Of interest, multiple antibiotic resistance profiles have been reported for *Aeromonas* and *Vibrio* [65,66,67,68].

In our study we report the results obtained concerning the resistome, mobilome and virulome analysis of 12 MDR marine bacterial strains by WGS, providing new insights on the genes involved in the AMR mechanisms.

The presence of class C and class D beta-lactamases is already reported for *S. algae* but few data are available for *V. anguillarum* and *V. parahaemolyticus*. In this study, we detected a chromosome-encoded AmpC in the *V. anguillarum* 28AD isolate, and a class A beta-lactamase (CARB-19) in *V. parahaemolyticus* VPE116 conferring resistance to penicillins and cephalosporins (first- and second-generation), and to amino- and carboxy-penicillins, respectively. Interestingly, the presence of *pqiAB* genes in our *S. algae* strains near the *bla*_AmpC_ gene and of *cusAB* genes near the *bla*_CARB-19_ in *V. parahaemolyticus* VPE116 may suggest the important role of heavy metal resistance-encoding genes in inducing the beta-lactam resistant phenotype in these marine bacteria.

Fluoroquinolones are frequently used in aquaculture. Some pentapeptide proteins belonging to the Qnr group were detected in the analyzed strains. Although these genes are generally located in plasmids (PMQR, except for QnrVC-like proteins), they were here detected in the bacterial chromosome both in *Shewanella* and *Vibrio* genera as already reported by Poirel and coauthors [69]. In fact, the origin of qnrA-like determinants was identified as being the chromosome of *S. algae* [8]. Then, their association to plasmids has allowed their spread in other bacterial species. It is noted that PMQR determinants generally confer only low-level quinolone resistance that alone does not exceed the clinical breakpoint but they can favor the selection of additional resistance mechanisms [70].

Of interest is the presence of some *S. algae* strains which carry the QnrA7 and belong to a group of strains isolated from the same aquaculture center while the strains harboring the QnrA3 protein have been obtained in other different sites and fish farms. Moreover, the *qnrVC6* gene, usually found in *V. cholerae*, was here detected in *V. anguillarum* 28AD. This is not really surprising, since *qnrVC*-like genes are currently spreading in other *Vibrio* species and Gram negatives such as *Pseudomonas aeruginosa*, *Pseudomonas putida* and *Citrobacter freundii* [71,72]. Interestingly, although the Qnr-like genes should provide resistance to quinolones, the phenotypes of all our strains are susceptible to ciprofloxacin and nalidixic acid, the two fluoroquinolones tested for.

Conversely, all the *S. algae* strains provided a resistant phenotype to fosfomycin, but the WGS analysis failed to detect the genes involved. This is probably due to the fact that they are still unknown and not reported on databases. Considering the data reported in literature, very high MICs were observed by Torri and coworkers [73] in clinical *S. algae* isolates during an Italian survey on shewanella-associated infections. This is probably an intrinsic resistance present in *S. algae* strains whose gene may be chromosomal-encoded considering a few presence of plasmids in these strains.

Regarding tetracycline resistance, only vibrios harbored *tet34-* or/and *tet35*-encoding genes. In particular, the *tet34* gene has been found in some Gram-negative genera (*Pseudomonas*, *Serratia* and *Vibrio*) and is unique to environmental bacteria [74]. Surprisingly, two *S. algae* strains, 38LV and 57CP, were resistant to tigecycline, a recent commercialized antibiotic used against MDR pathogens to treat hospital-acquired pneumonia, ventilator-associated pneumonia and other infections [75]. Tigecycline is scarcely investigated in aquaculture because it is mainly used in human medicine. The mechanism of resistance is still unknown, probably due to the presence of efflux pumps widely detected by WGS.

Globally, the data obtained for the resistome study seem to be in line with that reported for Italian aquaculture [64,65,66,67,68].

The mobilome analysis has been particularly difficult to perform, because of the Illumina technology which uses a mating pair and short read sequencing. Despite some difficulties, we identified integrative and conjugal elements in the 353M strain isolated in the open sea. This is an interesting isolate considering its scarce relation to aquaculture strains but potentially able to behave as an ARG reservoir in the same environment facilitating the spread of these elements and of their genes carried on to other marine bacteria or human pathogens. Furthermore, the presence of large size elements of ICE associated in *S. algae* strains allow us to highlight the contribution of these elements in aquaculture that, to the best of our knowledge, has never been investigated. The role of the detected bacteriophages has yet to be clarified. However, their involvement in the horizontal gene transfer of ARGs that flank these elements during the excision process, packing these bacterial ARGs in the mature virion particle and disseminating them in the genome of a new host cell, is not to be excluded. A scheme of how ARGs transfer occurs among bacteria is described in Figure 3.

Interestingly, both *S. algae* and *V. parahaemolyticus* VPE116 strains were shown to be prone to infection by bacteriophages belonging to different genera of marine and human pathogenic bacteria. Other transposable elements such as IS family transposases have been investigated. They are intracellular MGEs which need to be integrated in other mobile elements such as plasmids, bacteriophages or ICEs. Most of these ISs did not carry passenger genes such as transcription regulators, methyltransferases and antibiotic resistance, only the isolate 38LV carried a MFS transporter and an AraC family transcriptional regulator downstream the IS*481* family transposase, whereas the isolates 82CP and 353M carried a LysR and an AraC family transcriptional regulators downstream the transposase genes, respectively.

## 5. Conclusions

In conclusion, NGS technology is a useful means of detecting ARGs, MGEs and virulence factors although its reliability is dependent upon the availability of updated databases. Moreover, the genotype analysis sometimes does not correspond to the phenotype as in our case for some classes of antibiotics (fosfomycin and quinolones). Thus, the results obtained from this technology can act as a potential warning about the resistance of a microorganism. However, although transcriptome or proteome analysis is more adequate in detecting the real presence of the products generated by the predicted ARGs, WGS can help in analyzing the surrounding of the ARGs, the regulators involved during the transcription and possible mobilization of the MGEs. Here, we only analyzed the presence of these elements suggesting their possible involvement in the final MDR phenotype of our strains. Further studies should be performed to confirm how the MGEs act in the aquatic microbial community and if the genetic transfer occurs but this is not the scope of the current work. Among the data obtained, particularly interesting are those deriving from the mobilome analysis of these three species (*S. algae*, *V. parahaemolyticus* and *V. anguillarum*). The presence of several bacteriophages, large ICEs and ISs of different families has allowed us to decipher the players that might have a crucial role in spreading ARGs and virulence factors in the aquatic environment.

## Figures and Tables

**Figure 1 microorganisms-08-00572-f001:**
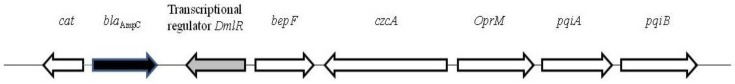
Genetic surrounding of the *bla*_AmpC_ gene found in the *S. algae* strains. The black arrow represents the *bla*_AmpC_ gene, the grey one the transcriptional regulator, whereas the white ones the other genes found in this structure.

**Figure 2 microorganisms-08-00572-f002:**
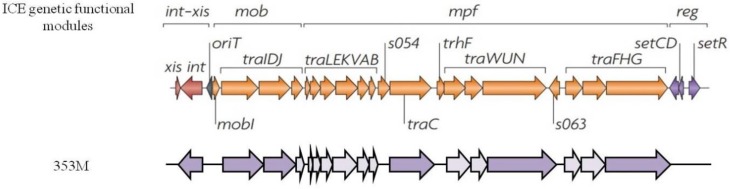
Description of the integrative and conjugative element (ICE) detected in the isolate 353M. At the top of the picture the genetic functional modules present in ICEs are reported [58]; at the bottom the encoding genes found in the strain 353M (dark violet) belonging to *xis/int*, *mob* and *mpf* modules and the other conjugal proteins (light violet) predicted as hypothetical but containing protein domains involved in conjugation and transfer at BLAST analysis are shown.

**Figure 3 microorganisms-08-00572-f003:**
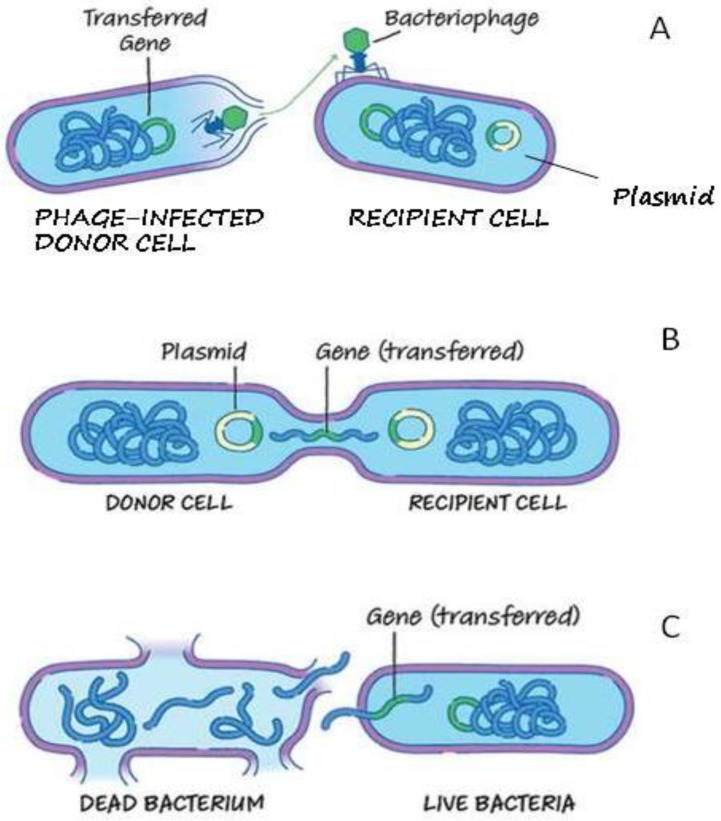
Scheme of the mechanisms of antimicrobial resistance genes (ARGs) transfer that are usually involved in bacterial cells: transduction, conjugation and transformation [76]. These mechanisms can occur also in the aquatic environment among autochthonous bacteria (aquatic bacterial species) and human pathogenic species that are temporally present in the aquatic environment. In the transduction process (**A**), a mature bacteriophage can move bacterial (virulence, antimicrobial resistance (AR), metabolism) genes to a new host cell. These genes can also be captured in a plasmid and spread by conjugation. In the mechanism of conjugation (**B**), a plasmid containing integrons, ISs, or transposons carrying ARGs can be transferred by ssDNA intermediate from a donor to a recipient cell. Also ICEs can use this mechanism followed by integration in the bacterial chromosome. During the transformation (**C**), the bacterial lysis can release DNA encoding for ARGs in the aquatic environment. This exogenous DNA can be captured from natural competent bacteria and integrated in their chromosomes.

**Table 1 microorganisms-08-00572-t001:** List of multidrug-resistant (MDR) marine isolates under study. Sampling dates, sources, species identification and antimicrobial resistance profiles are reported for each strain.

Strain	Sampling Date	Source	16S rRNA Gene Identification	Antimicrobial Resistance Profile
353M	10/08/2010	Water Open sea	*Shewanella algae*	CEF, FOX, IPM, AMX, CS, SUL, FOS
178CP	13/06/2011	Water Veneto II station	*Shewanella algae*	CEF, FOX, IPM, AMX, CS, SUL, FOS
146bCP	13/06/2011	Water Veneto II station	*Shewanella algae*	CEF, FOX, IPM, AMX, CS, SUL, FOS
144bCP	13/06/2011	Water Veneto II station	*Shewanella algae*	CEF, FOX, IPM, AMX, CS, SUL, FOS
219VB	30/09/2010	Water Veneto I station	*Shewanella algae*	CEF, FOX, IPM, AMX, CS, SUL, FOS
82CP	15/04/2011	Water Veneto II station	*Shewanella algae*	CEF, FOX, IPM, CS, SUL, FOS
38LV	22/07/2011	Water Varano lake	*Shewanella algae*	CEF, FOX, IPM, AMX, CS, SUL, FOS, TET, TGC
57CP	15/04/2011	Water Veneto II station	*Shewanella algae*	CEF, FOX, IPM, CS, SUL, FOS, TET, TGC
60CP	15/04/2011	Water Veneto II station	*Shewanella algae*	CEF, FOX, IPM, AMX, CS, SUL, FOS
83CP	15/04/2011	Water Veneto II station	*Shewanella algae*	CEF, FOX, IPM, AMX, CS, SUL, FOS
28AD	01/02/2007	European seabass	*Vibrio anguillarum*	CEF, FOX, AMX, TIC, AMC, TCC, TEM, CS, SUL, AKN
VPE116	26/06/2007	Water Caleri lagoon	*Vibrio parahaemolyticus*	AMX, TIC, TMN, KMN, AKN, GMI, CS, SUL

AMC, amoxicillin-clavulanic acid; AMX, amoxicillin; AKN, amikacin; CEF, cephalothin; CS, colistin; CTX, cefotaxime; FOS, fosfomycin; FOX, cefoxitin; GMI, gentamicin; IPM, imipenem; KMN, kanamycin; SUL, sulfonamides; TCC, ticarcillin-clavulanic acid; TEM, temocillin; TET, tetracycline; TGC, tigecycline; TIC, ticarcillin; TMN, tobramycin.

**Table 2 microorganisms-08-00572-t002:** List of resistance genes found in *S. algae* and *Vibrio* spp. grouped on the basis of their mechanism and antimicrobial target.

Antimicrobial	Pentapeptide Protein	Enzyme	Efflux System	Pump Family	Strain
**Beta-lactams**		OXA-55-like			All *S. algae*
		AmpC			All *S. algae, V. anguillarum* 28AD
			MexAB-OprM	RND	All *S. algae, V. parahaemolyticus* VPE116
**Fluoroquinolones**	QnrA3				353M, 219VB, 38LV, 57CP, 60CP
	QnrA7				144bCP, 178CP, 146bCP, 82CP, 83CP
	QnrVC6				*V. anguillarum* 28AD
			MdtK	MATE	All *S. algae, V. anguillarum* 28AD
			EmrAB-TolC	MFS	All *S. algae, V. parahaemolyticus* VPE116
	MepAB				All *S. algae, V. anguillarum* 28AD
			NorM	MATE	*V. anguillarum* 28AD, *V. parahaemolyticus* VPE116
			Bmr3	MFS	*V. anguillarum* 28AD
			MdtH	MFS	*V. anguillarum* 28AD
	MfpA				*V. anguillarum* 28AD
**Multiple substrates**			AcrAB-TolC	RND	All *S. algae, V. parahaemolyticus* VPE116
			AcrEF-TolC	RND	All *S. algae, V. parahaemolyticus* VPE116
			MdtABC-TolC	RND	All strains
			EmrD	MFS	All strains
			EmrYK-TolC	MFS	All strains excepting for 353M, 219VB, 82CP, 38LV
			MdtN	RND	All strains
			DrrA	ABC	All strains excepting for *V. anguillarum* 28AD
			Stp	MFS	All *S. algae*
			Bcr	MFS	All *S. algae*
			MdtG	MFS	*V. parahaemolyticus* VPE116
			MdlB	ABC	*V. parahaemolyticus* VPE116
**Tetracyclines**			TetR	MFS	*V. parahaemolyticus* VPE116
			Tet34	MFS	*V. anguillarum* 28AD, *V. parahaemolyticus* VPE116
			Tet35	MFS	*V. parahaemolyticus* VPE116
**Macrolides**			MacAB-TolC	ABC	All strains
			MdtE-TolC	RND	*V. anguillarum* 28AD, *V. parahaemolyticus* VPE116
**Polymyxins**		EptA			All strains
**Chloramphenicol**			MdtL	MFS	All strains
		Cat			All strains excepting for *V. anguillarum* 28AD

ABC, ATP binding cassette; MATE, multidrug and toxic compound extrusion; MFS, major facilitator system; RND, resistance-nodulation-cell division.

**Table 3 microorganisms-08-00572-t003:** Summary of mobile genetic elements found in *S. algae* and *Vibrio* spp. genomes.

Strain	Insertion Sequence (IS) Family	IS Family Subgroup	Integron/Integrase	Bacteriophage
**353M**	IS*630*, ISL*3*, IS*3* IS*630* IS*3* IS*4*	IS*Spu8* IS*Slo2* IS*Vvu3*	*IntI* Class I integron	Prophage-1 incomplete
**219VB**	IS*200*/IS*605*, IS*630*, ISL*3* IS*481* IS*110*, IS*3* IS*630* IS*4*	IS*Spu18* IS*Spu8* IS*Vvu3*	Class I integron	*Aeromonas* phage phiO18P (NC_009542)
**144BCP**	ISL*3* IS*481* IS*630*/IS*Spu8* IS*4* IS*3*, IS*630*, IS*110* IS*200*/IS*605* IS*110*, IS*4*	IS*Spu18* IS*Vvu3* IS*Sod* IS*Sba*	Class I integron	*Escherichia* phage D108 (NC_013594) *Shewanella* sp. phage 1/44 (NC_025463) Prophage-1 and Prophage-2 incomplete
**178CP**	ISL*3* IS*481* IS*630*/IS*Spu8* IS*4* IS*3*, IS*630*, IS*110* IS*200*/IS*605* IS*110*, IS*4*	IS*Spu18* IS*Vvu3* IS*Sod* IS*Sba*	Class I integron	*Escherichia* phage D108 (NC_013594) *Shewanella* sp. phage 1/44 (NC_025463) Prophage-1 and Prophage-2 incomplete
**82CP**	IS*630*, ISL*3*, IS*481*/IS*Spu18*, IS*110*, IS*3* IS*630*	IS*Sod16*	*IntI* Class I integron	Prophage-1 *Aeromonas* phage phiO18P (NC_009542)
**146BCP**	ISL*3* IS*481* IS*630*/IS*Spu8* IS*4* IS*3*, IS*630*, IS*110* IS*200*/IS*605* IS*110*, IS*4*	IS*Spu18* IS*Vvu3* IS*Sod* IS*Sba*	Class I integron	*Escherichia* phage D108 (NC_013594) *Shewanella* sp. phage 1/44 (NC_025463) Prophage-1 and Prophage-2 incomplete
**83CP**	IS*200*/IS*605*, ISL*3*, IS*110* IS*630* IS*630* IS*4* IS*3*	IS*Spu8* IS*Vvu3* IS*Sba6* IS*Slo2*	*IntI* Class I integron	Prophage-1 *Aeromonas* phage phiO18P (NC_009542)
**38LV**	ISL*3* IS*481* IS*110* IS*630*	IS*Spu18* IS*Sod16*	*IntI* Class I integron	-
**57CP**	IS*200*/IS*605*, ISL*3*, IS*110* IS*3* IS*630* IS*630* IS*4*	IS*Spu8* IS*Sod16* IS*Vvu3* IS*Sba6*	*IntI* Class I integron	*Enterobacteria* phage phi92 (NC_023693) incomplete
**60CP**	IS*200*/IS*605*, ISL*3*, IS*110* IS*3* IS*630*	IS*Slo2* IS*Sod16*	Class I integron	2 Unknown Prophages incomplete
**28AD**	IS*200*/IS*605*, IS*5*/IS*1182*, IS*As1* IS*30*	IS*Spu13* IS*Va6*	*IntI* Class I integron	2 Unknown Prophages incomplete
**VPE116**	IS*5*/IS*1182* IS*5* IS5 IS*3*	IS*Spu14* IS*Vpa3* IS*Visp3*	*IntI*	*Enterobacterial* phage mEp213 (NC_019720) *Vibrio* phage VP882 (NC_009016)

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
