# Peer review of "Resistome, Mobilome and Virulome Analysis of Shewanella algae and Vibrio spp. Strains Isolated in Italian Aquaculture Centers"

_microorganisms, 2020, doi:10.3390/microorganisms8040572_

Round 1

Reviewer 1 Report

The authors analyzed the whole genome sequence of MDR Shewanella and Vibrio strains isolated from aquatic places. Based on WGS data, they suggested that these strains carries some of genes probably responsible for drug resistance including genes that encode drug efflux pumps. They also found chromosomal mobile elements and genes that probably contribute to production of bacteriocin and hemolysin. This work provides some of insights into general questions how the MDR marine bacteria have acquired drug resistance in aquatic environment. However, it is highly speculative literature.

  1. The authors suggested involvement of genes encoding drug efflux pumps in MDR phenotype in addition to drug degradation and modification enzymes. This hypothesis is not agreeable. Some of drug efflux pumps encoded by chromosomal genes contribute to natural tolerance in bacteria while another ones are “silent” due to very low expression. If some of these efflux pumps are “over-produced” probably due to any mutations in their promoters, the host may acquire drug resistance. Therefore, the authors should show direct evidence that these drug efflux pumps “actually” are associated with MDR phenotype in these strains. In addition, most of drug efflux pumps sound like homologue to those of E. coli and P. aeruginosa that are well-studied bacteria species. However, most of them have not been characterized in Shewanella and Do these pumps export similar drugs to those of E. coli and P. aeruginosa?

  1. As similar issues, are type I, II, III, IV and VI secretion systems functional in these bacteria? As well as hemolycin assays, production and/or activity of these secretion proteins should be tested. Are some of genes that probably contribute to drug resistance and virulence transferrable via the mobile elements? To justify the suggestion, it should be also tested.

  1. Scientific writing and English quality must be improved.

Author Response

R1: The authors suggested involvement of genes encoding drug efflux pumps in MDR phenotype in addition to drug degradation and modification enzymes. This hypothesis is not agreeable. Some of drug efflux pumps encoded by chromosomal genes contribute to natural tolerance in bacteria while another ones are “silent” due to very low expression. If some of these efflux pumps are “over-produced” probably due to any mutations in their promoters, the host may acquire drug resistance. Therefore, the authors should show direct evidence that these drug efflux pumps “actually” are associated with MDR phenotype in these strains. 

A1: Thank you for your comment. As you mentioned, drug efflux pumps has a minor role in establishing a MDR phenotype. This is essentially due to the high affinity of an enzyme for its substrate, when expressed, if compared to pumps (they can export a high range of different molecules outside the cell membrane).

However, in this study, the WGS can only reveal the presence of enzymes able to degrade or modify a specific substrate and some other mechanisms of resistance. We tried to give an explanation of the results obtained both from the antimicrobial susceptibility testing and the presence of the genes revealed by WGS. Considering tetracyclines, for example, the presence of tet34 and tet35 genes may explain a small inhibition zone for this class of antibiotics in the susceptibility tests. Conversely, for other classes such as beta-lactams, the beta-lactamases found (OXA-55, AmpC and CARB-19) are undoubtedly more efficient than efflux pumps that could contribute in a minor way to the final phenotype. In this study we propose to consider the possible contribution of efflux pumps as mechanism of resistance in the marine bacteria when there is no evidence of a modifying or degrading enzyme causing the resistant phenotype. However, this is only a proposed explanation. In the discussion and conclusions these statements are now better described highlighting that it is only an hypothesis.

R1: In addition, most of drug efflux pumps sound like homologue to those of E. coli and P. aeruginosa that are well-studied bacteria species. However, most of them have not been characterized in Shewanellaand Do these pumps export similar drugs to those of E. coli and P. aeruginosa? 

A1: The drug efflux pumps found in these MDR marine strains belong to the same families of those found in Gram negatives (ABC, MATE, RND and MFS). These pumps are generally more studied in E. coliP. aeruginosa and other Gram negative bacteria of clinical interest. It seems that they can contribute to export the same molecules. In Shewanella algae strains, we, in fact, found for example, the presence of Mex-type pumps that are involved in beta-lactam resistance also in P. aeruginosa. It is clear that more studies should be performed to determine the real contribution of these pumps in these two species, for example, through an RNA-seq analysis, however this aspect is out of  the scope of this study.  

R2: As similar issues, are type I, II, III, IV and VI secretion systems functional in these bacteria? As well as hemolycin assays, production and/or activity of these secretion proteins should be tested. Are some of genes that probably contribute to drug resistance and virulence transferrable via the mobile elements? To justify the suggestion, it should be also tested. 

A2: The aim of the study was neither to explore the pathogenicity, virulence and antibiotic resistance of the strains isolated in aquaculture nor to demonstrate the transfer of the genetic elements involved. Our goal was to explore the presence, in marine strains mainly of veterinarian interest, of genes and genetic elements that are known to be important in strains isolated from humans. The expression of these genes and their transferability to human/clinical strains are only a possibility that however deserve to be taken into consideration as a reservoir of virulence and antibiotic resistance. In fact, this possibility has been already extensively demonstrated in scientific studies. Our point is then just to show that the aquaculture strains are bearing elements and genes potentially of medical interest in that already detected in clinical strains..

This has now been explained in more detail and the aims more focused in the discussion. 

R3: Scientific writing and English quality must be improved. 

A3: We revised and  improved the English quality of the manuscript.

Reviewer 2 Report

The manuscript by Vanessa Zago et al. describes about the charactereistics of isolated multidrug resistant Shewanella algae and Vivrio spp. through draft genome sequencing analysis. Authors has showed that all resistant genes for each antibiotic, and also showed an evidence of mobile element and virulence factors from draft genome data. It may be possible to miss some of information, because authors just use draft genome data, not complete genome data. Even though, all contents in this manuscript were well-described and well-written. Overall, it would be said that a quality of result and explanation is sufficient enough for the acceptance as very good quality publication.

Author Response

R1: The manuscript by Vanessa Zago et al. describes about the charactereistics of isolated multidrug resistant Shewanella algae and Vivrio spp. through draft genome sequencing analysis. Authors has showed that all resistant genes for each antibiotic, and also showed an evidence of mobile element and virulence factors from draft genome data. It may be possible to miss some of information, because authors just use draft genome data, not complete genome data. Even though, all contents in this manuscript were well-described and well-written. Overall, it would be said that a quality of result and explanation is sufficient enough for the acceptance as very good quality publication. 

A1: Thanks for your comment. We are glad that the reviewer has found our manuscript interesting in the field.  

Reviewer 3 Report

Dear authors,

Interint paper, thanks

Here my suggestions:

1.INTRODUCTION:

I miss a short description about the Vibrio spps likewise the authors have done about Shewanella.

3.RESULTS

Please for Table 1, make differences among each individual strains, something like a colour or a line to separate each cell.

4. DISCUSSION 

5. CONCLUSION

I am missing a link with a possible transmission of ARGs and/or IMAGEs from these environment straight to clinical strains. What do the authors think? Could they add a sentence about this possible transmission, share or even cycle among environmental-clinical strains (see paper last chapter and figure https://doi.org/10.3390/app9122486)

Also I would like to suggest to add a schedule or drawn summarizing the manuscript and also a paper.

Thanks

Author Response

R1: INTRODUCTION: I miss a short description about the Vibrio spps likewise the authors have done about Shewanella. 

A1: We have done it. We have improved the introduction regarding the state of the art for vibrios. 

R2: 3.RESULTS: Please for Table 1, make differences among each individual strains, something like a colour or a line to separate each cell. 

A2: We have introduced a line in order to separate the cells and to make more understandable the data reported for each strain. 

R3: I am missing a link with a possible transmission of ARGs and/or IMAGEs from these environment straight to clinical strains. What do the authors think? Could they add a sentence about this possible transmission, share or even cycle among environmental-clinical strains (see paper last chapter and figure https://doi.org/10.3390/app9122486). Also I would like to suggest to add a schedule or drawn summarizing the manuscript and also a paper. Thanks 

A3: The data reported in this manuscript have highlighted the presence of some MGEs deriving from other aquatic and pathogenic bacteria. In particular the WGS analysis has revealed high similarity of the ICE tra genes, found in a Shewanella algae strain, to other deposited on Genbank database belonging to Salmonella enterica and in less manner to Pseudoalteromonas spp. Regarding bacteriophages, we found phages that usually can infect other aquatic bacteria such as Aeromonas spp. or pathogenic bacteria such as E. coli or other Enterobacteria. They are able to integrate both in the genome of Shewanella algae and Vibrio spp. isolates here analyzed. There is evidence, in literature, of the ability to transfer these MGEs at intraspecies and interspecies level so this could be a way through which the aquatic bacteria can move possible new ARGs to human pathogens, temporally living in the same environment, but also human pathogens can use phages to transfer ARGs that are important in the clinical settings to the aquatic bacteria contributing to maintain the aquatic resistome. We added in the manuscript a scheme describing the mechanisms of ARGs transfer that can occur among aquatic microbiota. 

Round 2

Reviewer 1 Report

I am aware of the author’s suggestion. However, most of statements suggested in this manuscript including the title are misleading. This work shows definitely neither “Resistome”, “mobilome” nor “virulome” analysis because there is no direct evidence that most of genes found in the WGS indeed are responsible for drug resistance, mobilization and virulence phenotypes in strains used for this work. A main point of this work just showed draft genome sequences in some of multidrug resistant (MDR) Shewanella and Vibrio pathogens, then identified some of candidate genes responsible for MDR, and also subsets of genes and sequence elements that are probably associated with virulence and mobilization.

Some of genes encoding drug modification/degradation enzymes would closely contribute to MDR phenotype in highly possible according to resistance profile for each strain while it is still unclear whether most of genes including virulence and mobilization-related genes are functional because these phenotypes were not shown (in case that their expression may be very low level or there may be some SNPs affecting their function). The authors should carefully discuss the limitation of this work, then reconstruct the manuscript.